# Notalgia Paresthetica Review: Update on Presentation, Pathophysiology, and Treatment

Christopher Robinson [1], Evan Downs [2,†], Yanet De la Caridad Gomez [3,†], Chinonso Nduaguba [1], Parker Woolley [1], Giustino Varrassi [4], Jatinder Gill [1], Thomas T. Simopoulos [1], Omar Viswanath [2,5] and Cyrus A. Yazdi [1,*]

1   Beth Israel Deaconess Medical Center, Department of Anesthesiology, Critical Care and Pain Medicine, Harvard Medical School, Boston, MA 02215, USA
2   School of Medicine, Louisiana State University Health Sciences Center, Shreveport, LA 71103, USA
3   Pain Management Center, Harvard Medical School, Boston, MA 02215, USA
4   Paolo Procacci Foundation, 00193 Roma, Italy
5   Department of Anesthesiology, Creighton University School of Medicine, Omaha, NE 68124, USA
*   Correspondence: cyazdi@bidmc.harvard.edu
†   These authors contributed equally to this work.

**Abstract:** Purpose of Review: Notalgia paresthetica (NP) is a chronic cutaneous neuropathy primarily characterized by localized pruritus and associated dysesthesias, including sensations of pain, numbness, and tingling. The sensory neuropathy characteristic of NP is thought to result from spinal nerve entrapment caused by degenerative changes in the spine or musculoskeletal compression. This review summarizes the current medical literature with a focus on the past five years regarding NP, its pathophysiology, presentation, and current treatment options. Recent Findings: Though treatments exist with varying efficacy, to date, there exists no definitive treatment for NP. Treatment options for NP are varied and range from topical and oral agents to interventional procedures and physical therapy. Of the treatments evaluated, topical capsaicin remains the most efficacious treatment for NP. Conclusions: The lack of established treatment guidelines makes treating NP complicated as it dramatically affects patients' quality of life. Further research with larger sample sizes is needed to evaluate better the most effective treatment and dosing regimen for patients afflicted with NP.

**Keywords:** nostalgia paresthetica; cutaneous neuropathy; topical treatments; capsaicin





## 1. Introduction

Notalgia paresthetica (NP), first described in 1934, is a chronic cutaneous neuropathy primarily characterized by localized pruritus and associated dysesthesias, including sensations of pain, numbness, and tingling [1,2]. The symptoms of NP are typically unilateral and located medial or inferior to the scapula within the middle or upper back [3,4]. The symptomatic area may be associated with a hyperpigmented patch, most often secondary to chronic scratching and rubbing to relieve the discomfort (Figure 1) [2]. The symptoms usually last for years, and the course of the disease is characterized by periods of remission and exacerbation [2,4–6].

Patients with NP are most commonly women over 40 years of age [1,4,7]. Although a cross-sectional study linked the female gender to worse disease severity, statistical analyses from two more recent studies did not find any correlation between gender and severity [1,2,8]. The sensory neuropathy characteristic of NP is thought to result from spinal nerve entrapment caused by degenerative changes in the spine or musculoskeletal compression. Treatment options for NP are varied and range from topical and oral agents to interventional procedures and physical therapy [9]. The lack of established treatment guidelines makes treating NP difficult and significantly affects patients' quality of life [10]. This review aims to

summarize the current medical literature with a focus on the past five years regarding NP, its pathophysiology, presentation, and current treatment options.

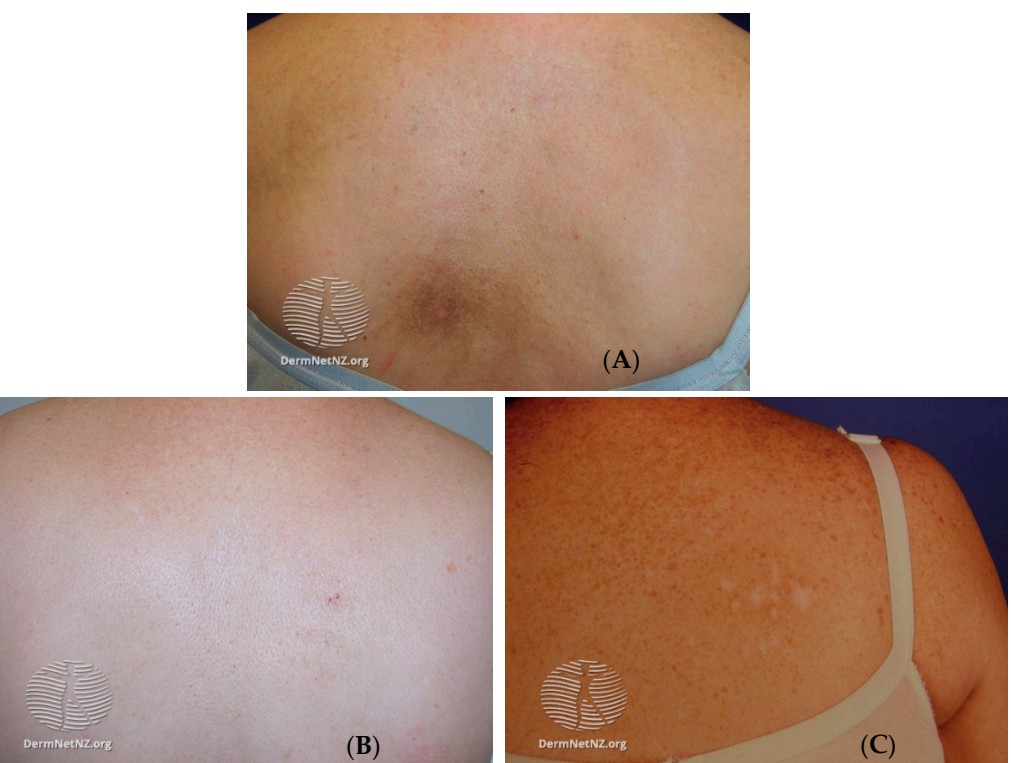

**Figure 1.** The dermatological sequelae from the scratching in notalgia paresthetica. (**A**). Hyperpigmentation. (**B**). Scratch marks. (**C**). Scarring from excessive scratching. Images provided by Dermnet.

## 2. Epidemiology/Risk Factors

NP most commonly occurs in middle-aged patients, with an incidence of two to three times higher in women than men [4,11]. Moreover, women may carry an increased risk of developing a more severe form of NP. However, other studies have shown no correlation [12]. Although the mean age of patients with NP is between 50–60, NP can occur in people as young as six years of age [4,13].

While the etiology of NP is not yet fully understood, symptoms may originate from sensory neuropathy [13]. A cross-sectional study found that patients diagnosed with acquired NP were more likely to have degenerative cervical and thoracic spine disease than those with back pain without pruritus [2]. Additionally, lesions were identified in the vertebrae corresponding to affected skin dermatomes in NP patients [3]. This further lends support to the theory of nerve impingement in the pathogenesis of acquired cases. In younger patients with NP, an association with multiple endocrine neoplasia 2B (MEN-2B) has been identified and is thought to be inherited in an autosomal dominant pattern; these patients were found to have no evidence of cervical or thoracic disease or a history of traumatic injury to the spine [4,11]. Furthermore, elevated body mass index predisposes one to a longer disease course but does not increase the risk of developing NP [13]. In contrast, temperature change has been identified as a potential exacerbating factor [14].

## 3. Pathophysiology

The exact pathophysiology of NP remains unknown. One leading hypothesis is that damage to the posterior cutaneous branches of T2–T6 spinal nerves leads to a thoracic polyradiculopathy and contributes to the pathogenesis of NP; damage to these spinal nerves is thought to occur from impingement caused by degenerative changes in the spine or by musculoskeletal compression [1,2,4,6,7]. This proposed mechanism is supported

by findings demonstrating a significant overlap between the dermatomal distribution of NP and the location of spinal pathology, including degenerative changes and herniated disks, observed radiographically [6,7]. A more recent study, however, found the overlap between NP location and radiographic findings to correlate in only 16% of individuals studied, suggesting spinal abnormalities may not completely explain the pathogenesis of NP [15]. A cross-sectional study of 45 patients with NP and 35 patients without NP found a significantly higher number of herniated disks on the C6–C7 segment in the NP group than in the control group, suggesting cervical spine degenerative changes, and not just thoracic lesions, which may also contribute to NP; interestingly, radiological findings in the thoracic spine were similar between the NP group and the control group [2]. Although patients with NP commonly have spinal lesions, the relationship between spinal pathology and NP has yet to be investigated.

In addition to degenerative changes in the spine, musculoskeletal nerve compression, including compression by muscle fibers, has also been postulated in the pathophysiology of NP [4,7,16]. The sensory nerve branches at T2-T6 emerge from the multifidus spinae muscle at a right angle which makes them more susceptible to compression and injury compared to nerves elsewhere in the back (Figure 2) [16]. A recent prospective cohort study of 117 patients with NP corroborated that T2–T6 dermatomes were the most commonly symptomatic dermatomes, with T3 found to be the most involved one of all [1]. Supporting this view, strengthening and stretching exercises targeting the mid-to-upper back have been shown to reduce symptoms of NP by relieving stretching and/or compression of the affected nerves [17,18].

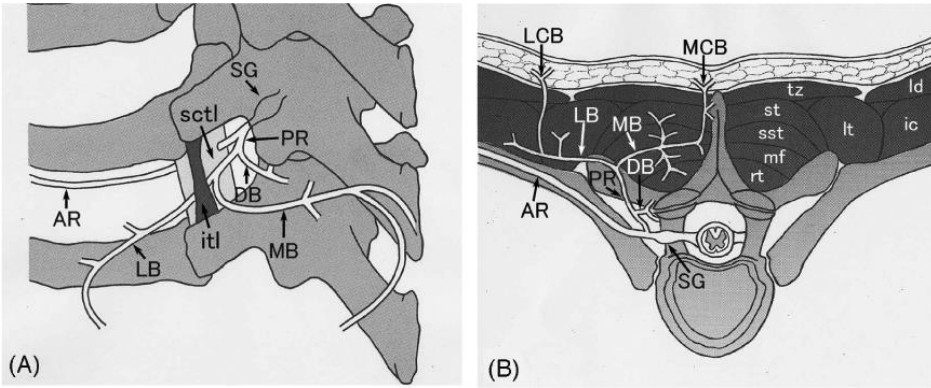

**Figure 2.** Illustration demonstrating the configuration of the posterior rami in the posterolateral (**A**) and axial (**B**) views. Note the cutaneous branches branching off at roughly right angles in the thoracic region. Abbreviations: AR—anterior ramus; DB—descending branch; LCB—lateral cutaneous branch; LB—lateral branch; MCB—medial cutaneous branch; MB—medial branch; ic—iliocostalis; itl—intertransverse ligament; ld—latissimus dorsi; lt—longissimus thoracis; mf—multifidus; rt—rotatores; sctl—superior costotransverse ligament; sg—spinal ganglion; sst - semispinalis thoracis; st—spinalis thoracis; tz—trapezius. Illustration provided by Ishizuka [19].

A possible increase in dermal innervation has also been implicated in the pathogenesis of NP [20]. However, a subsequent study biopsied 14 patients with NP and did not find any change in the number of nerves within lesions compared to non-lesional, healthy skin [21]. Instead, more recent evidence suggests that a decrease rather than an increase in dermal innervation may be associated with NP. A retrospective study of 65 patients with NP analyzed skin biopsies of 21 patients with NP for intraepidermal nerve fiber density and found a significantly decreased number of nerve fibers in the lesional area compared to the non-lesional area [15]. These results point to a peripheral cause of sensory neuropathy in NP.

Reports of hereditary forms of NP are rare but raise the question of whether genetic predisposition could also contribute to the pathogenesis of NP. NP has been previously

reported in cases of multiple endocrine neoplasia type 2A (MEN2a) [15,22]. However, the lack of systemic manifestations in most patients with NP makes MEN2a an unlikely cause.

## 4. Presentation and Diagnosis

NP is a sensory neuropathy marked by sensory changes and chronic pruritus leading to lichenification and dark pigmentation. NP is an often overlooked and underdiagnosed disease, with the diagnosis being based on history and physical examination. Primary skin lesions have not been reported in the literature, as dermatological findings have been attributed to the sequelae of chronic scratching (Figure 1). Pigment deposition in the affected area is a result of post-inflammatory changes [13].

Patients typically present with symptoms of pruritus or paresthesia located medial or inferior to the scapula, typically in the T2–T6 dermatomes (Figures 2 and 3) [11]. Other sensory changes include pain, change in temperature, paresthesia, hypoesthesia, or hyperesthesia [13]. Symptoms usually are unilateral but bilateral manifestations have been reported [9]. NP is most commonly found on the left side of the body, opposite the dominant hand, most likely due to the majority of the population being right-hand dominant [23]. The true incidence and prevalence of NP are not fully known but are believed to be underreported and underdiagnosed, as skin findings can resemble other chronic pruritic conditions. At the same time, biopsy results are not specific [13]. Though not needed, a biopsy can be helpful when ruling out other diagnoses. Biopsy of the pigmented lesions may demonstrate hyperkeratosis, macrophages with intracellular melanin accumulation, and necrotic epidermal keratinocytes [14,24]. NP should be suspected in patients who present with chronic pruritus and/or paresthesias without underlying dermatologic pathology, especially if there is evidence of cervical or thoracic spine pathology, as patients with NP had a statistically significant higher prevalence of nerve compression, nucleus pulposus extrusion, or degenerative spine changes [2].

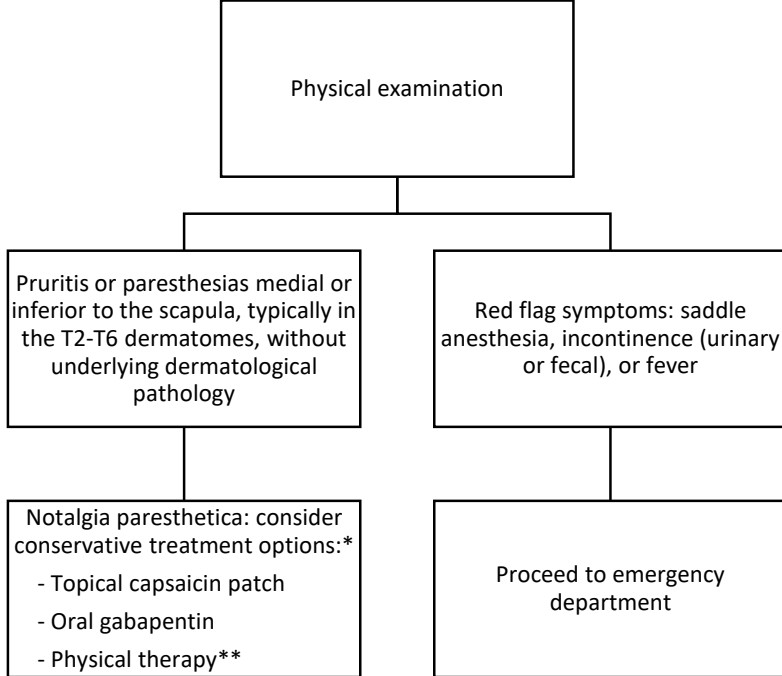

**Figure 3.** Flow chart detailing the findings in notalgia paresthetica, including conservative management. * There are several conservative treatment options with varying efficacy; representative examples are given as there are no guidelines for treatment. ** Physical therapy can be a supplemental option, especially for patients affected by atrophied paraspinal muscles or who report a shoulder with a reduced range of motion.

## 5. Differential Diagnosis

The differential diagnoses for NP include pigmented contact dermatitis, patchy parapsoriasis, lichen simplex chronicus, macular amyloidosis, and tinea versicolor—amongst others. NP presents similarly to these diagnoses; however, many of these conditions are diagnosed by a dermatologist after a punch biopsy. Pigmented contact dermatitis, also known as Riehl melanosis, is typically a result of an acquired allergy to a fragrance or cosmetic product and commonly presents on the face; patch testing can be completed to differentiate between NP and pigmented contact dermatitis [25].

Parapsoriasis is a difficult diagnosis often made by a dermatologist based on clinical history. A punch biopsy will demonstrate nonspecific findings such as linear parakeratosis, acanthosis, and perivascular lymphohistiocytic infiltration [26]. There will be multifocal lesions associated with parapsoriasis in contrast to NP with distinct posterior lesions in a dermatomal [26].

Lichen simplex chronicus is a form of neurodermatitis that presents with hypertrophied skin secondary to chronic itching, similarly seen in NP [27]. The most distinct difference between NP and lichen simplex chronicus is that the latter is believed to be secondary to psychological stressors, with chronic itching being an emotional coping mechanism [28]. The plaques that develop in lichen simplex chronicus occur in more accessible areas of the body, such as the head, neck, hands, arms, and genitals [28]. Similar to NP, a biopsy is not necessary for lichen simplex chronicus but will reveal post-inflammatory changes such as parakeratosis, thickened epidermal rete, acanthosis, spongiosis, and perivascular along with interstitial inflammation secondary to chronic scratching of the affected area [28].

Macular amyloidosis presents in a similar distribution as NP, most commonly between the scapulas and associated with pruritus; however, on gross examination, it appears as a rippled pattern [29,30]. If clinical suspicion is high for macular amyloidosis, a punch biopsy can be performed demonstrating amyloidosis on Congo red staining in contrast to the post-inflammatory changes seen in NP [31–33].

Tinea versicolor, or pityriasis versicolor, is a cutaneous fungal infection caused by the *Malassezia spp*. It can be easily differentiated from NP using potassium hydroxide and examination under a Wood's lamp highlighting the areas of fungal infection; a biopsy is not required, but fungal filaments can be seen on microscopy [34,35].

## 6. Treatment

Despite very few studies having been performed to assess the clinical efficacy of treatment options for NP, pharmacological treatments do exist with varying levels of success (Table 1) [36]. The effectiveness of most treatments has only been determined in isolated studies with relatively few subjects involved. One such case study involved intravenous (IV) lidocaine to treat NP after oral antihistamines and topical hydrocortisone cream was deemed ineffective [37]. Three infusions of lidocaine were given at two-week intervals to one patient with a dosage of 1 mg/kg bolus followed by a 4 mg/kg infusion administered over one hour [37]. Up to a 90% reduction in pruritis was achieved following the third and final infusion, but symptoms gradually returned to baseline over the following month, and no adverse side effects related to the infusions were reported [37]. Though lidocaine demonstrated potential as a treatment for NP, maintenance infusions every 3–4 weeks would be required to maintain symptom alleviation, and studies with larger numbers of participants must be completed in order to assess IV lidocaine's long-term efficacy and safety as a treatment for NP [37].

**Table 1.** Clinical Efficacy of Treatments for Notalgia Paresthetica.

| Author and Year | Groups Studied and Intervention | Results and Findings | Conclusions |
|---|---|---|---|
| Chtompel et al. 2017 [37] | One patient underwent IV lidocaine infusions. | Itch reduction in patient immediately following infusions with symptoms returning to baseline within one month. | Treatment was effective over the short term; however, larger studies and evaluation of long-term efficacy and safety are required. |
| Mülkoğlu et al. 2020 [2] | Forty-five patients were treated with local intradermal lidocaine injections. | Significant reduction in pain and pruritis in patients lasting one to three months post injection. | Effective in the short term, but long-term efficacy and safety remain undetermined. |
| Andersen et al. 2016 [5] | Three patients were topically treated with 8% capsaicin patches. | Patients reported immediate symptom relief with duration varying from a few days to three months. | Clinical efficacy undetermined due to wide range of results. Further studies with a larger patient pool are necessary. |
| Leibsohn et al. 1992 [38] | Twenty-four patients were treated with 0.025% topical capsaicin patches for four months. | Seventy percent of patients achieved some degree of symptom relief but returned to baseline after cessation of therapy. | One of the first treatments for NP and relatively effective in some patients. Long-term effects beyond four months are unknown. |
| Maciel et al. 2014 [14] | Twenty patients divided into two groups were compared-one group treated with 0.025% topical capsaicin, and the second treated with oral 300 mg gabapentin. | Only gabapentin treatment group reported significant symptom reduction. | Gabapentin was well tolerated and effective. |
| Goulden et al.1998 [39] | One patient was treated with bilateral paravertebral block using bupivacaine (0.75%) and 40 mg methylprednisone. | Patient had completer resolution of symptoms for 12 months. | Effective treatment of NP, though limited by sample size and expertise required to perform blocks. |
| Savk et al. 2001 [40] | Four patients were treated with oral 300–900 mg oxcarbazepine twice a day. | Three of four patients reported improvement at one and six months of therapy. | Oxcarbazepine was well tolerated, with only one patient experiencing headache and dizziness. |
| Ochi et al. 2016 [41] | Seven patients were treated with 0.1% topical tacrolimus for six weeks. | Patients experienced decrease in mean itch score (0 to 10) from 6.6 +/− 1.9 to 4.6 +/− 2.1 ($p < 0.02$). | Topical treatment reduced pruritis intensity and/or frequency with return of symptoms after cessation. |
| Yeo et al. 2013 [42] | One patient was treated with oral 10 mg amitriptyline daily for three months. | Reduction of pruritis scores from 7/10 to 4/10 with sustained relief after discontinuation for one month. | Oral amitriptyline was effective in treating one patient without side effects. |
| Poterucha et al. 2013 [43] | Two patients were treated with amitriptyline/ketamine topical cream. | One of two patients experienced partial relief of pruritis with no relief in the second. | Topical amitriptyline/ketamine cream provided partial relief of pruritis. |
| Weinfeld et al. 2007 [44] | Two patients were treated with botulinum toxin A. First patient received 16 units, and second patient received 24 units, followed by a second dose of 48 units. | One patient had complete resolution after one injection for 18 months, and second patient required two injections to achieve complete resolution. | Botulinum toxin A injection produced resolution of NP, but dosing regimen is variable. |

**Table 1.** *Cont.*

| Author and Year | Groups Studied and Intervention | Results and Findings | Conclusions |
|---|---|---|---|
| Perez-Perez et al. 2014 [45] | Five patients were treated with 48–56 units of botulinum toxin A. | Three patients had partial improvement of symptoms, and two had worsening of symptoms. | Variable and partial improvement with injection of botulinum toxin A. |
| Maari et al. 2014 [46] | Double-blinded randomized controlled trial consisting of 20 patients receiving either a mean dose of 142 units of botulinum toxin A or saline. | No significant difference was observed between the botulinum toxin A and saline group. | Botulinum toxin A was ineffective in treating NP. |
| Perez-Perez et al. 2010 [47] | Five patients received narrow-band UVB radiation with an average of 32.8 sessions with a mean cumulative dose of 33.75 J/cm$^2$. | All five patients had reductions in symptoms, and two patients had complete resolution. | Narrow-band UVB radiation resulted in reduction of symptoms. |
| Fleischer et al. 2011 [17] | Two patients were treated with physical therapy. | Both patients achieved significant improvement in symptoms after strengthening and stretching exercises targeting scapular and pectoral muscles. | Physical therapy is an effective treatment in patients affected by atrophied paraspinal muscles or who report a shoulder with a reduced range of motion. |
| Savk et al. 2007 [48] | Fifteen patients were treated with TENS over 10 sessions. | Transient pruritis relief was achieved in some patients, but symptoms returned to baseline upon the cessation of treatment. | May be effective in some symptom relief and should be considered as part of a multi-modal therapy approach in treating NP. |
| Williams et al. 2010 [49] | One patient underwent surgical decompression. | Symptom relief was achieved in this patient postoperatively. | Surgical decompression successfully resulted in symptom relief. Further studies are required before implementation as a treatment for NP due to its invasiveness. |

One of the first treatments developed for NP was low- and high-dose topical capsaicin patches. Topical capsaicin was evaluated in a case series involving three patients treated with an 8% capsaicin patch for one hour with varying results [5]. All patients reported immediate itch relief, but the duration of this relief ranged from two days to three months [5]. Low-dose capsaicin patches of 0.025% concentration were evaluated in a four-month study using a group of 24 patients with NP [38]. Seventy percent of the patients reported pruritis relief with up to a 90% reduction in symptoms [38]. The majority of patients returned to baseline upon the cessation of therapy, but the long-term side effects of this therapy remain relatively unknown [38].

Though not long-lasting, intradermal lidocaine injections have been shown to be effective. The effects of local intradermal injections of lidocaine into affected dermatomes were studied on a group of 45 patients with NP [2]. Three separate injections spaced two weeks apart were performed, depositing one milliliter of 0.33% lidocaine in 1 cm intervals along the C2–T6 spinous processes [2]. The patient's pain and pruritis were determined utilizing the visual analog scale (VAS). Significant reductions in both pain as assessed by the visual analog scale (VAS) and pruritis scores were achieved with symptom improvement lasting up to three months [2].

Oral pharmacological options include the use of gabapentin. Daily oral 300 mg gabapentin was compared against daily 0.025% topical capsaicin patches for four weeks in 20 patients [18]. Ten patients were assigned to each treatment group, and the effects of each

treatment were determined using the VAS scale [18]. The use of 300 mg daily gabapentin resulted in a significant reduction in itching intensity, while capsaicin patches did not result in significant symptom reduction [18]. The side effects of gabapentin were well tolerated within the treatment group, and further investigation of gabapentin as a treatment option utilizing double-blind, randomized controlled studies is warranted to determine its efficacy as a long-term treatment for patients with NP [18].

Other pharmacological options include corticosteroids, oxcarbazepine, tacrolimus, amitriptyline, a combination of topical amitriptyline/ketamine cream, and intradermal botulinum toxin A with limited efficacy and reproducibility [9,13]. Topical corticosteroids offer little benefit in the treatment of NP, but one case report demonstrated 12 months of relief when a paravertebral block was performed by injecting bupivacaine (0.75%) and 40 mg of methylprednisone [39,50]. Despite the relief, it was only found in one patient with no further follow-up studies. The widespread applicability may be limited due to the expertise required [39]. In a small pilot study consisting of four patients, administration of 300–900 mg of oral oxcarbazepine twice a day resulted in a decrease in pruritis and pain in three of the four patients, with only one patient experiencing side effects [40]. Similar to other studies, no follow-up studies with a larger group and control were performed, limiting the use of oxcarbazepine.

Topical 0.1% tacrolimus applied twice a day resulted in pruritis improvement in six of seven patients at six weeks with a decrease in mean itch score (0 to 10) from $6.6 +/- 1.9$ to $4.6 +/- 2.1$ ($p < 0.02$); unfortunately, symptoms returned to baseline with cessation of therapy [41]. Oral 10 mg amitriptyline daily for three months resulted in a decrease in pruritis (7/10 to 4/10) and sustained relief without side effects even after discontinuation at one month [42]. When used topically and combined with ketamine, it demonstrated partial relief in one of two patients with no improvement in the second [43].

Intradermal botulinum toxin A has produced inconsistent results in the treatment of NP. Sixteen units were injected in one patient with resolution of symptoms for up to 18 months, and a second patient also had resolution but required a second dose (48 units) after the first dose of 24 units [44]. Five patients were treated with a higher initial dose of 48–56 units, with three patients having partial improvement and two having worsening symptoms [45]. In a double-blinded randomized controlled trial consisting of 20 patients receiving either a mean dose of 142 units of botulinum toxin A or saline, no statistically significant difference was observed between the two groups [46].

Non-pharmacological treatments for NP have been assessed with varying levels of efficacy. Narrow-band ultraviolet B (UVB)was used to treat five patients, with symptom reduction in all five patients and complete resolution in two patients [47]. No side effects were experienced in any of the five patients, and the two with resolution did not have the return of symptoms six months after discontinuation [47]. Of note, patients with MEN-2A may have symptomatic improvement with UV exposure [13].

Physical therapy to strengthen paraspinal muscles (iliocostalis, longissimus, and spinalis muscles) and pectoral muscles have shown effectiveness in a subset of patients either affected by atrophied paraspinal muscles or who report a shoulder with a reduced range of motion [17,51]. The response to physical therapy lends further credence to the theory that nerve impingement may be responsible for the development of NP [9,11,12].

Transcutaneous electrical nerve stimulation (TENS) was evaluated as a treatment for NP in a group of 15 patients [48]. All patients received ten treatment sessions in the affected area, each session with a 20 min duration at a frequency of 50–100 Hz [48]. Patients did have significant pruritis relief but only transiently [48]. TENS and physical therapy should be considered non-pharmacological treatments that can be added to pharmacological interventions as part of a multi-modal approach to NP treatment [48]. Additionally, surgical decompression of the affected nerve root has been used successfully for lasting relief in NP, but the first case reporting the use of decompression was in 2009, with no known supporting cases reported since that time [49].

## 7. Prognosis and Complications

Although NP is not fatal, it can negatively impact the quality of patients' lives. Complications of NP are related to chronic scratching, which results in skin excoriations, increasing the risk of infections and the development of hyperpigmentation and lichenification. Unfortunately, there is no definitive treatment for NP, with treatments varying widely in efficacy. Symptoms may last for numerous years, resolving and returning. The mechanism for the spontaneous resolution of symptoms is not entirely understood, as symptoms can regress without treatment or recur without any evident inciting factors [12].

## 8. Conclusions

NP is a sensory neuropathy characterized by chronic pruritus and paresthesias in a circumscribed region of the mid-to-upper back. The affected area may develop post-inflammatory hyperpigmentation appearing as a hyperpigmented patch, most commonly along the T2–T6 dermatomes. Its diverse presentation makes NP challenging to diagnose; therefore, it is an underrecognized condition. Although widely mistaken to be a dermatologic disease, currently available evidence suggests that dermatologic complications result from underlying nerve pathology, which can be caused by various mechanisms, including nerve compression by degenerative changes of the spine and other musculoskeletal structures nearby. Given the multiple possible causes of NP, the diagnosis and management of this disease require an interdisciplinary approach. Even with various treatment modalities available, NP has been shown to impact the daily lives of patients. Thus significantly, future efforts in the treatment of NP should aim to elucidate the most effective multi-modal approach for minimizing the symptoms of NP and improving patients' quality of life.

**Author Contributions:** Conceptualization, C.R., C.N., P.W., J.G., T.T.S. and C.A.Y.; methodology, C.R. and E.D.; validation, E.D. and Y.D.l.C.G.; formal analysis, O.V.; investigation, C.R., C.N., P.W., J.G., T.T.S.; resources, O.V.; data curation, C.R., C.N., P.W., J.G., T.T.S.; writing—original draft preparation, C.R., C.N., P.W., J.G., T.T.S.; writing—review and editing, O.V., C.A.Y. and G.V. All authors have read and agreed to the published version of the manuscript.

**Funding:** This research did not receive any external funding.

**Institutional Review Board Statement:** Not applicable.

**Informed Consent Statement:** Not applicable.

**Data Availability Statement:** All the data are available by the corresponding author on reasonable request.

**Acknowledgments:** The authors are grateful to the Paolo Procacci Foundation for the support in the publishing process.

**Conflicts of Interest:** All the Authors declare no conflict of interest.

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
