# Peer review of "Notalgia Paresthetica Review: Update on Presentation, Pathophysiology, and Treatment"

_clinpract, doi:10.3390/clinpract13010029_

Round 1

Reviewer 1 Report

The paper entitled “Notalgia Paresthetica Review: Update on Presentation, Pathophysiology, and Treatment” is a review based on the description, incidence, and clinical significance notalgia paresthetica (NP). This disorder is a chronic neuropathy characterized by pruritus, dysesthesias, sensations of pain, numbness, and tingling. Although little is known, it is thought to be caused by nerve entrapment caused by degenerative changes in the spine or musculoskeletal compression.

The manuscript nicely summarizes the literature regarding NP, its pathophysiology, clinical presentation, and current treatment options. Although no standardized treatments and guidelines exist, therapeutic options include topical and oral agents, interventional procedures, and physical therapy.

Notalgia can be a debilitating disorder, thus proper clinical understanding of these manifestations is important in the differential diagnosis. The authors correctly remind clinicians of the importance of signs, symptoms, and differential diagnoses of nostalgia. Management of these patients involves education, counseling, and procedures that can be noninvasive or invasive.

The authors should consider a step-by-step flow chart that can be useful for clinicians when addressing these patients, indicating first and second-level examinations with appropriate timing for follow-up assessments and signs and symptoms that warrant urgent neurologic evaluations.

The main question of the manuscript was to provide a review of the description and clinical manifestations of notalgia. The study has been correctly planned. It is well well-written and of clinical interest. The study provides objective results and is relevant in this field. The manuscript adds to the current literature. The conclusions are consistent with the continents presented throughout the text and the main questions have been addressed in an appropriate manner. References are appropriate and up to date.

Author Response

We thank you for considering our manuscript entitled “Notalgia Paresthetica Review: Update on Presentation, Pathophysiology, and Treatment.”
We have implemented your feedback including adding the figures and tables at the end of the manuscript and have addressed each of the reviewers comments with responses in red below.

The authors should consider a step-by-step flow chart that can be useful for clinicians when addressing these patients, indicating first and second-level examinations with appropriate timing for follow-up assessments and signs and symptoms that warrant urgent neurologic evaluations.

o We have added the flowchart as a separate chart and referenced it on Line 133 as Figure 3

Reviewer 2 Report

The authors have performed an interesting review about notalgia paresthetica. I have the following suggestions.

In the Abstract, Summary should be replaced with Conclusion.

Line 57 - "the most recent medical literature" - please specify the period.

The authors should specify what type of review they performed.

I consider that the Epidemiology/risk factors Section should be moved before the Pathophysiology Section.

In the Conclusions section, the authors should present the conclusions of the article and should not include bibliographic references.

Spaces are missing between words in certain sentences. The authors should revise this throught the manuscript.

Author Response

We thank you for considering our manuscript entitled “Notalgia Paresthetica Review: Update on Presentation, Pathophysiology, and Treatment.”
We have implemented your feedback including adding the figures and tables at the end of the manuscript and have addressed each of the reviewers comments with responses in red below.

Reviewer 2: • In the Abstract, Summary should be replaced with Conclusion.

o Line 33: Summary has been replaced with conclusion. • Line 57 - "the most recent medical literature" - please specify the period. o Line 57-58: We have specified “the current medical literature with a focus on the past five years”. • The authors should specify what type of review they performed.

o Line 1: We have specified that it is a systematic review. • I consider that the Epidemiology/risk factors Section should be moved before the Pathophysiology Section.

o Line 59: We have moved the Epidemiology/risk factor section to Line 59 in front of the Pathophysiology section.
• In the Conclusions section, the authors should present the conclusions of the article and should not include bibliographic references.

o We have removed the bibliographic references from the conclusion section. • Spaces are missing between words in certain sentences. The authors should revise this through the manuscript.

o We have updated the manuscript throughout the remove the extraneous spaces.